# *LTBP4*, *SPP1*, and *CD40* Variants: Genetic Modifiers of Duchenne Muscular Dystrophy Analyzed in Serbian Patients

**DOI:** 10.3390/genes13081385

**Published:** 2022-08-04

**Authors:** Ana Kosac, Jovan Pesovic, Lana Radenkovic, Milos Brkusanin, Nemanja Radovanovic, Marina Djurisic, Danijela Radivojevic, Jelena Mladenovic, Slavica Ostojic, Gordana Kovacevic, Ruzica Kravljanac, Dusanka Savic Pavicevic, Vedrana Milic Rasic

**Affiliations:** 1Department of Neurology, Clinic of Neurology and Psychiatry for Children and Youth, 11000 Belgrade, Serbia; 2Centre for Human Molecular Genetics, Faculty of Biology, University of Belgrade, 11000 Belgrade, Serbia; 3Laboratory of Medical Genetics, Mother and Child Health Care Institute of Serbia “Dr Vukan Cupic”, 11000 Belgrade, Serbia; 4Department of Neurology, Mother and Child Health Care Institute of Serbia “Dr Vukan Cupic”, 11000 Belgrade, Serbia; 5Faculty of Medicine, University of Belgrade, 11000 Belgrade, Serbia

**Keywords:** Duchenne muscular dystrophy, single-nucleotide polymorphisms, *LTBP4*, *SPP1*, *CD40*

## Abstract

Background: Clinical course variability in Duchenne muscular dystrophy (DMD) is partially explained by the mutation location in the *DMD* gene and variants in modifier genes. We assessed the effect of the *SPP1*, *CD40*, and *LTBP4* genes and *DMD* mutation location on loss of ambulation (LoA). Methods: SNPs in *SPP1*-rs28357094, *LTBP4*-rs2303729, rs1131620, rs1051303, rs10880, and *CD40*-rs1883832 were genotyped, and their effect was assessed by survival and hierarchical cluster analysis. Results: Patients on glucocorticoid corticosteroid (GC) therapy experienced LoA one year later (*p* = 0.04). The modifying effect of *SPP1* and *CD40* variants, as well as *LTBP4* haplotypes, was not observed using a log-rank test and multivariant Cox regression analysis. Cluster analysis revealed two subgroups with statistical trends in differences in age at LoA. Almost all patients in the cluster with later LoA had the protective IAAM *LTBP4* haplotype and statistically significantly fewer *CD40* genotypes with harmful T allele and “distal” DMD mutations. Conclusions: The modifying effect of *SPP1*, *CD40*, and *LTBP4* was not replicated in Serbian patients, although our cohort was comparable in terms of its *DMD* mutation type distribution, SNP allele frequencies, and GC-positive effect with other European cohorts. Cluster analysis may be able to identify patient subgroups carrying a combination of the genetic variants that modify LoA.

## 1. Introduction

Duchenne muscular dystrophy (DMD) is the most common muscular dystrophy and has a progressive devastating course. It is caused by a complete deficiency of protein dystrophin due to loss-of-function mutations in the *DMD* gene. The clinical course of the disease is usually clearly defined as follows: muscle weaknesses leading to loss of independent ambulation (LoA) by the teenage years, followed by the loss of the hand function and unavoidable complete immobility. Cardiomyopathy and respiratory failure progress in the advanced phases of the disease, leading to the lethal outcome. The multidisciplinary care approach has prolonged life expectancy in DMD patients [1,2,3,4,5]. However, variability in the disease progression is documented, and we are now seeing patients with a prolonged period of mobility, delayed onset of cardiomyopathy, and need for respiratory support [6,7,8,9]. Duchenne muscular dystrophy phenotype variability is influenced by standards of care, therapy with glucocorticoid corticosteroids (GCs), and the genetic architecture of the patient, including the type and location of DMD mutations and genetic modifiers [10,11,12,13].

Loss of ambulation is the most obvious and easily measurable disease progression milestone in DMD patients. It is used for studying the natural history of the disease or comparing any of the interventions applied, from everyday care to innovative medications. Loss of walking ability is a consequence of chronic inflammation, regeneration failure, and fibrosis in the muscles. These processes result from a cascade of events triggered by the loss of sarcolemma integrity due to dystrophin deficiency [14]. The only symptomatic therapy used for all DMD patients is GCs, which slow the progression of muscle weakness [15,16]. Median age at LoA is 10 years in GC-naïve patients and 13 years in GC-treated patients [16]. Longer GC therapy has a cumulative effect, lasting for more than 1 year and delaying disease progression milestones for 2.1 to 4.4 years compared to treatment duration which lasts for less than 1 month [17]. Glucocorticoid corticosteroids achieve this effect by binding to the regulator of inflammation, NF-κB (nuclear factor kappa-light-chain-enhancer), in activated B cells, and by decreasing immune cell proliferation via cell-cycle arrest, possibly through NFAT5 (nuclear factor of activated T-cells 5) [18].

Some out-of-frame mutations in the *DMD* gene allow dystrophin expression in traces, and those patients express a milder phenotype [19]. In patients with the deletion of exons 3 to 7, the median age at LoA was up to 15 years, while, in those with mutations eligible for the skipping of exon 44, median age at LoA was 14.8 years [20]. The latter group mostly includes patients with the isolated deletion of exon 45.

The *SPP1* (secreted phosphoprotein 1) gene was the first described DMD modifier gene. It codes for osteopontin, a cytokine involved in processes of regeneration, inflammation, and tumor progression [21]. Osteopontin is overexpressed in dystrophin-deficient muscles [22], where it upregulates collagen expression in fibroblasts by promoting TGF-β signaling through the induction of MMP9 protease [23]. The minor allele G of rs28357094, a single-nucleotide polymorphism (SNP) located in the *SPP1* promoter, was associated with earlier LoA [21,24]. In an in vitro luciferase assay, the G allele downregulated promoter activity [25]. Although reduced *SPP1* transcription was detected in DMD patients carrying one or two G alleles, no significant difference in osteopontin level was detected. On the other hand, patients with the G allele showed a reduced number of CD68^+^ macrophage infiltrating cells [11,26], which suggests that there is a more complex mechanism underlying the modifying effect of rs28357094. This SNP was also challenged as a pharmacodynamic marker for GC response, rather than being a direct gene modifier of DMD [10].

The *LTBP4* (latent transforming growth factor-β binding protein 4) gene has also been described as a DMD modifier [27]. Its IAAM haplotypes of rs2303729–rs1131620–rs1051303–rs10880 (specifically the T allele of rs10880) were associated with later LoA. The homozygous IAAM haplotype was shown to delay mean age at LoA for almost 2 years in GC treated patients, and almost for 1.5 years in GC-naïve patients [27]. The LTBP4 protein is an aspect of TGF-β signaling with a role in fibrosis [28]. The protective effect of IAAM haplotype is related to the resistance of the latent TGF-β complex to proteolysis, leading to decreased TGF-β signaling, a reduction in sarcolemma permeability, and fibrosis [11]. Mdx mice with the protective *Ltbp4* allele (with 36 bp insertion) and mdx mice expressing the human *LTBP4* gene are characterized by an increased retention of TGF-β within the latent complex and a better outcome regarding fibrosis and phenotype characteristics in comparison to mdx controls [29,30].

The *CD40* gene encodes TNFRSF5 (tumor necrosis factor receptor superfamily member 5), which is an important costimulatory protein expressed on antigen-presenting cells with a role in the activation of T cells that further triggers the nuclear factor κB (NF-κB) pathway in a broad range of biological processes [11]. The role of *CD40* in dystrophin-deficient muscles is insufficiently clear, but it is known that T-cell depletion modulates fibrosis and response to GC therapy. The minor T allele of rs1883832 in *CD40* gene has been shown to be associated with a 1 year earlier LoA [31]. Although the T allele reduces *CD40* transcriptional activity in vitro, an increased level of the *CD40* transcript and decreased levels of the CD40 protein were detected in a patient’s muscle biopsy with the minor T allele [11].

In this study, we assessed the effect of variants in *SSP1*, *LTBP*4, and *CD4*0 genes on age at LoA in Serbian DMD patients from the referral clinical registries taking into account the GC treatment applied, as well as *DMD* mutation type and location. Since the studied DMD modifier genes are involved in pathological processes of muscle inflammation, degeneration, and fibrosis [11], we performed cluster analysis in order to identify more homogenous groups of patients with regard to LoA and genetic data.

## 2. Materials and Methods

### 2.1. Subjects and Criteria

Unrelated patients were selected from the clinical registers from two University Clinics in Serbia, Clinic for Neurology and Psychiatry for Children and Youth, and Mother and Child Health Care Institute of Serbia “Dr Vukan Cupic”. The inclusion criteria were DMD genetically confirmed by MLPA or *DMD* gene sequencing, accurate information on GC treatments, and the availability of historical DNA samples used for genetic diagnosis. Priority was given to non-ambulant patients, bearing in mind that the age at LoA was the outcome measure in this study. LoA was defined as inability to walk 10 m independently and was estimated to the nearest half a year of age. Patients were considered as “GC-treated” if they had received steroid therapy (prednisolone or deflazacort daily) for at least 1 year before losing ambulation. Three patients with the deletion of exon 45 and one patient with the deletion of exons 3 to 7 were excluded from the study, since it is known that these mutations are associated with prolonged median age at LoA [20,32,33,34]. According to these criteria, a total of 95 DMD patients were selected. On the basis of the location of mutations in the *DMD* gene, the patients were divided into two groups: “proximal” if the mutation was located upstream of intron 44, affecting long dystrophin isoforms Dp427 and Dp260 only, and “distal” if the mutation encompassed intron 44 and regions downstream of it, affecting one or more shorter isoforms (Dp140, Dp116, and Dp71), in addition to the longer ones [13].

### 2.2. SNP Genotyping

The genotyping analysis conducted for variants rs28357094 in the *SPP1* gene, rs2303729, rs1131620, rs1051303, and rs10880 in the *LTBP4* gene, and rs1883832 in the *CD40* gene was performed via an allelic discrimination assay using appropriate TaqMan assays (C___1840809_10, C__22271866_10, C___8714829_10, C___8714838_20, C___2936821_1_, and C__11655919_20, respectively, ThermoFisher Scientific, Waltham, MA, USA). The PCR mixture contained 10–20 ng of genomic DNA, 1× FastGene^®^ Probe (Nippon Genetics Europe GmbH, Düren, Germany), 0.25 μM ROX additive (Nippon Genetics Europe GmbH, Düren, Germany), 0.6 mg/mL BSA (New England Biolabs, Ipswich, MA, USA), and 0.6× appropriate TaqMan^®^ SNP Genotyping Assay. The reaction was performed in The StepOnePlus™ Real-Time PCR System (ThermoFisher Scientific, Waltham, MA, USA) with the following temperature profile: 2 min at 95 °C, followed by 40 cycles of 5 s at 95 °C and 30 s at 60 °C. The profile also included pre-PCR read and post-PCR read steps, both performed for 30 s at 60 °C. The fluorescence was measured using the StepOne Software v2.3. ROX dye was used as a passive internal reference. Ten randomly selected samples were analyzed in duplicate for each assay with 100% concordance.

### 2.3. Statistical Analysis

Analyzed SNPs were tested for Hardy–Weinberg equilibrium using the HardyWeinberg package in R. To examine the effect of different factors (GC therapy, SNP genotypes, and mutation location in *DMD* gene) on age at LoA, survival analyses were implemented with the ambulatory patients being censored. To assess the effect of one factor on LoA, median age at LoA was estimated and compared between different groups of patients using Kaplan–Meier curves and the log-rank test. For *SPP1* and *CD40* variants, a dominant model was used for the corresponding minor alleles, as previously described [21,31]. The *LTBP4* variants were not considered separately; instead, the haplotypes were reconstructed using the haplo.stats package in R. In addition to the most frequent haplotypes (the referent VTTT and alternative IAAM), all other haplotypes had individual frequencies at or below 10% and were grouped with the VTTT haplotype as a single category in the analysis, referred to as other. For the *LTBP4* haplotypes, dominant, additive, and recessive models for IAAM haplotype were considered [27]. To assess the effect of examined variables on LoA simultaneously, and to provide the strength of effect for each individual variable LoA, multivariate Cox regression analysis was performed. The age at LoA was a dependent variable, while GC treatment, mutation location in *DMD* gene, and SNP genotypes, according to the dominant model, were used as covariates. Additional Cox regression analysis was performed with the same covariates but including two interaction terms—one between GC treatment and SNP genotypes and the other between GC treatment and mutation location.

To detect subtle patient categories that could further help to explain the variability in disease progression, cluster analysis was performed. Cluster analysis, which is a form of unsupervised machine learning, represents a set of multivariate statistical tools and algorithms designed to detect patterns in complex data and organize the information in relatively homogeneous groups [35]. Unlike the methods of classical statistics that provide the causal relationship between given variables, clustering is more directed toward describing the data through detecting patterns and regularities [36]. Cluster analysis was performed on 64 patients who have lost ambulation. The variables used were encoded in binary terms for simplicity using label encoding. *SPP1* and *CD40* SNPs, as well as the *LTBP4* haplotype, were encoded according to a dominant model. Age at LoA was transformed into binary format with respect to the median age at LoA, such that ages below and above 10.75 years were encoded as zeros and ones, respectively. This threshold was chosen as the median and mean, and the age at LoA of all 64 selected patients was similar (10.75 years and 10.78 years, respectively). The mutation location in *DMD* gene was encoded as either proximal or distal, whereas GC therapy was not included due to bias toward GC use (approximately 70% of patients, 45 out of 64 analyzed, were/are receiving GC therapy). Missing values were imputed using the “most frequent” strategy, where each missing value was replaced by the mode of variable distribution. The optimal number of clusters was estimated using both the elbow method and Ward’s linkage dendrogram. We opted for the hierarchical clustering algorithm, as it is generally regarded to work well on small datasets, it does not require previous knowledge of the number of clusters, and its results are easily reproducible, since it does not include a random seed component [35]. Clustering evaluation was assessed through domain knowledge, as well as by internal validation indices: C–H (Calinski–Harabasz) index, Dunn’s index (DI), Davies–Bouldin (DB) index, Silhouette index (SI), and SDbw validity index (S_Dbw). Fisher’s exact test of independence was used for the comparisons of categorical variables (proportion of reference vs. variant genotypes/haplotypes from corresponding dominant models, and proportion of proximal vs. distal mutations) between the obtained clusters. To compare age at LoA between the obtained clusters, the Wilcoxon–Mann–Whitney test was performed after testing data normality using the Shapiro–Wilk test. The clustering analysis and evaluation were performed in Python3 (3.9.0). 

Statistical analyses were performed in R ver. 4.0.4 (R Core Team, 2020) [37]. A two-tailed *p*-value at 0.05 was considered significant in all tests.

## 3. Results

### 3.1. Effect of Examined Individual Factors on LoA in DMD Patients

A total of 99 patients were selected. Four patients, either with the deletion of exons 3–7 or with the deletion of exon 45, were excluded from the study. Two non-ambulatory patients with the deletion of exon 45 lost mobility at the age of 10 and 13 years and did not use GC therapy. Two patients, one 7-year-old with the deletion of exon 45 and one 12-year-old with the deletion of exons 3–7, were unrestrictedly mobile and used GC therapy. Therefore, a total of 95 DMD patients (mean age 15.8 ± 7.2 years) were analyzed in this study. Among them, 64 (67.4%; mean age 18.7 ± 7.1 years) were wheelchair-dependent and 31 (32.6%; mean age 10 ± 2.2 year) were ambulant. The GC therapy was applied in 73 patients (76.84%). The mutation spectrum in the *DMD* gene was as follows: deletions were detected in 65 patients (68.4%), duplications were detected in nine patients (9.5%), and small mutations were detected in 21 patients (22.1%). The mutations were “proximal” in 40 patients (42.1%) and “distal” in 55 patients (57.9%).

Patients using GC therapy lost ambulation later than patients without GC therapy (mean age at LoA 11.14 and 9.95 years, respectively). This difference was statistically significant (*p* = 0.021) (Table 1, Figure 1A), confirming the protective effect of GC therapy on motor performance. When comparing the mean age at LoA in patients carrying “proximal” and “distal” *DMD* mutations (11.08 and 10.59 years, respectively), a statistical trend was observed (*p* = 0.093) (Table 1, Figure 1B). When observing only patients on GC therapy, there was not any difference in mean age at LoA between patients with “proximal” and “distal” mutations (11.17 and 11.13 years, respectively) (*p* = 0.46) (Table 1, Figure 1C). However, in the group of patients without GC therapy, those with “distal” mutations lost ambulation earlier than those with “proximal” (9.27 and 10.88 years, respectively) (*p* = 0.013) (Table 1, Figure 1D).

All tested SNPs showed no deviation from the Hardy–Weinberg equilibrium (rs28357094 *p* = 0.82; rs1883832 *p* = 0.52; rs2303729 *p* = 0.92; rs1131620 *p* = 0.86; rs1051303 *p* = 0.86; rs10880 *p* = 0.85). The minor allele frequency (MAF) for the studied SNPs was in accordance with the observed frequency for the European (non-Finnish) population from gnomAD genome database r3.0. The exception was minor allele for *CD40* rs1883832, showing a frequency of 0.330 in comparison to 0.276 in the gnomAD genome database r3.0.

From our cohort, 93 patients were successfully genotyped for the *SPP1* rs28357094. The TT genotype was observed in 55 patients, while the TG and GG genotypes were observed in 38. When stratifying non-ambulant patients (*n* = 62) according to the dominant model for minor allele G, patients who carried the TT genotype (*n* = 36) lost ambulation at a mean age of 10.94 years and patients with TG and GG genotypes (*n* = 26) lost ambulation at mean age of 10.62 years (*p* = 0.32) (Table 1, Appendix A). The examined *SPP1* genotypes had no effect on LoA as well when considering only GC-treated (*p* = 0.79) or GC-untreated patients (*p* = 0.69) (Table 1, Appendix A).

The *CD40* rs1883832 was genotyped in all 95 patients. Forty patients had the CC genotype, and 55 patients had the CT or TT genotype. Non-ambulant patients (*n* = 64) were stratified according to the dominant model for minor allele T. The median age at LoA in patients with the CC genotype was 10.69 years, while, in patients with the CT and TT genotypes, it was 10.88 years (*p* = 0.24) (Table 1, Appendix A). Similarly to the *SPP1* genotype, the examined *CD40* genotypes had no effect on LoA when considering only GC-treated (*p* = 0.45) or GC-untreated patients (*p* = 0.80) (Table 1, Appendix A).

The *LTBP4* haplotypes rs2303729, rs1131620, rs1051303, and rs10880 were reconstructed in 89 patients. The frequency distribution of the observed haplotypes was as follows: VTTT 52.8%, IAAM 31.5%, IAAT 10.1%, ITTT 3.9%, VAAM 1.1%, and VTTM 0.6%. We analyzed the effect of IAAM haplotype in 58 non-ambulant patients using dominant, additive, and recessive models. In patients without the IAAM haplotype (*n* = 27), the mean age at LoA was 10.39 years, while, in patients with at least one IAAM haplotype (*n* = 31), it was 10.69 years (*p* = 0.61) (dominant model, Table 1, Appendix A). In patients with one IAAM haplotype (*n* = 27), the mean age at LoA of 10.54 years was higher than in patients without IAAM haplotype (*n* = 26, 10.39 years) and lower than in patients carrying two IAAM haplotypes (*n* = 5, 11.5 years), but this difference was not statistically significant (*p* = 0.56) (additive model, Table 1, Appendix A). Patients homozygous for the IAAM haplotype (*n* = 5) lost ambulation at the mean age of 11.5 years, whereas patients with all other combinations of the examined haplotype lost ambulation at the mean age of 10.46 years (*p* = 0.29) (recessive model, Table 1, Appendix A). There was no difference in the mean age of LoA when stratifying patients according to IAAM haplotype and GC treatment by the dominant (Table 1, Appendix A), additive (Table 1, Appendix A and S4C), or recessive model (Table 1, Appendix A).

A multivariate Cox regression model was constructed with LoA as the outcome variable and GC treatment, mutation location in the *DMD* gene, *SPP1* genotype, *CD40* genotype, and *LTBP4* haplotype used as covariates (Table 2). The GC treatment was shown to have a protective effect (HR = 0.44, *p* = 0.01), while distal location of *DMD* mutation significantly increased the risk of LoA (HR = 1.92, *p* = 0.03). There were no significant effects of *SPP1* variant, *CD40* variant, and *LTBP4* haplotype on age at LoA (HR = 1.03, *p* = 0.9; HR = 0.86, *p* = 0.59; HR = 0.72, *p* = 0.25, respectively). The overall model showed a trend toward significance (*p* = 0.068, concordance = 0.64). In the Cox regression model that included interaction terms, a statistically significant interaction was observed between GC treatment and the distal location of *DMD* mutation (HR = 0.21, *p* = 0.02). On the other hand, the interaction term between GC treatment and genetic variants was not significant for *SPP1* variant (HR = 0.69, *p* = 0.58), *CD40* variant (HR = 0.68, *p* = 0.55), or *LTBP4* haplotype (HR = 1.25, *p* = 0.76). The overall model was at a borderline level of significance (*p* = 0.056, concordance = 0.68).

### 3.2. Cluster Analysis

The optimal number of clusters determined by the elbow method and the dendrogram implied the existence of either two or four clusters, with similar indices of internal measure (for two clusters: DB score = 1.056, Silhouette score = 0.409, S_Dbw score = 2.559, Calinski–Harabasz index = 47.894; for four clusters: DB score = 0.809, Silhouette score = 0.517, S_Dbw score = 2.422, Calinski–Harabasz index = 101.513) (Appendix A). However, since the entire dataset consisted of only 64 patients, the separation of patients into four clusters would surely have led into overfitting, as the number of instances in a single cluster would be rather small. For this reason, we performed subsequent cluster analysis on the two larger clusters—cluster I encompassing 41 patients and cluster II with 23 patients. The cluster profiles are given in Table 3. The obtained clusters showed no significant difference in the distribution of rs28357094 genotypes in the *SPP1* gene according to the dominant model (*p* = 1). However, there was a statistically significant difference in the distribution of the rs1883832 genotype in the *CD40* gene (*p* = 5.7 × 10^−4^) and *LTBP4* haplotypes (*p* = 2.8 × 10^−6^). Cluster II had five times fewer rs1883832 genotypes with the *CD40* harmful minor allele T, while the frequency of the protective the IAAM haplotype was much higher than expected (*p* = 2.8 × 10^−6^). Moreover, only a single case without IAAM haplotype was observed in this cluster. In addition, the obtained clusters differed significantly in the distribution of mutation location in the *DMD* gene (*p* = 0.02). The “distal” mutations were observed three times less frequently in cluster II, whereas both clusters had almost an equal number of “proximal” mutations. Lastly, cluster II had three times fewer patients who lost their ambulation earlier than the median age of 10.75. When comparing the distribution of actual age at LoA between the two clusters, a trend toward significance was observed (W = 339, *p* = 0.06). When considering all statistically significant variables from the cluster analysis together, we compared the patients who carried all harmful (rs1883832 CT/TT genotypes, other/other *LTBP4* haplotypes, and distal mutation location) traits with those with all protective factors (rs1883832 CC genotype, other/IAAM and IAAM/IAAM *LTBP4* haplotypes, and proximal mutation location). Interestingly, the former group consisted of eight patients from cluster I and the latter group consisted of eight patients from cluster II. In the group from cluster I, five out of eight patients lost their ambulation earlier than the median age of 10.75, while, in the group from cluster II, six out of eight patients lost their ambulation later.

## 4. Discussion

In this study, we assessed the effects of three modifier genes on the clinical course of DMD patients from Serbia, thus filling the gap regarding the lack of information on *DMD* genetic modifiers in populations from southeastern Europe.

Our cohort included 95 well-characterized, unrelated patients. They were genetically diagnosed with DMD, and their mutation type distribution in the *DMD* gene was comparable with that described in other populations [6,19,38,39]. Since LoA was used as the main disease progression milestone, we excluded patients with deletions of exons 3 to 7 and exon 45, as they are known to ameliorate disease progression [20,33,40]. Data on GC therapy were available for all patients, and they were considered to be “GC-treated” if they had received GC therapy for at least 1 year before losing mobility. Although the number of studied patients was comparable with that in some previously published studies [41], the sample size of our cohort was one of the limitations for this genetic association study.

According to our results, GC therapy prolonged the mean age of LoA for 1.2 years in DMD patients receiving GC therapy for 1 year or longer prior to LoA. The same criteria for GC therapy were used in a study of 440 patients showing a delay in median age at LoA of 2.1 to 4.4 years and a better outcome from the use of deflazacort [17]. In our study, many more patients were treated with prednisolone. Possible reasons for the absence of a greater effect could be that the therapy was introduced later in life or the heterogeneity among patients using GC, with not all taking the proposed dose of 0.75 mg/kg prednisolone or 0.9 mg/kg deflazacort, and with different durations of treatment (1 year and more) prior to LoA.

We observed a similar predominance of “distal” DMD mutations in our patients as in the Italian cohort and Cooperative International Neuromuscular Research Group—Duchenne Natural History Study (CINRG-DNHS) [10,12]. In our cohort, steroid-naïve patients with “distal” mutations lost ambulation earlier than those with proximal mutations, indicating the importance of timely GC treatment for all patients, especially for patients with distal mutations. In an Italian cohort, the effect of “distal” mutations on LoA was not examined, but the study pointed out distal mutation as a risk factor for worse results in pulmonary function tests. This finding was partially explained with questionable volition and efforts, having known that, in patients with distal mutations, even shorter dystrophin isoforms, present in the central nervous system, are affected [12,13,42].

The first study showing the modifying effect of rs28357094 in the *SPP1* gene was conducted on two DMD cohorts composed of 106 patients from the Padova cohort and 156 patients from the CINRG cohort. The rare G allele of rs28357094 (present in 35% of patients) was associated with a 1 year earlier mean age at LoA in the Padova cohort and weaker grip in CINRG cohort, among which the most evident findings were obtained in GC-treated non-ambulant patients [21]. We did not confirm a significant effect of the *SPP1* genotype on age at LoA. Similarly, an association was not observed between *SPP1* genotype and age at LoA in cohorts from five European centers (London, Newcastle, Montpellier, Leiden, and Ferrara) [41]. According to the frequency of 41% obtained for the rs28357094 G allele and lack of deviation from the Hardy–Weinberg equilibrium, our group is comparable to groups from the other studies, in which the G allele frequency ranged from 29% in Italian cohort to 47.7% in Leiden patients [10,21,24,41]. Although the abovementioned groups, including ours, encompass fewer than 100 patients, a lack of association between *SPP1* genotype and age at LoA was also described in 254 non-ambulant patients from the United Dystrophinopathy Project cohort [27]. In patients who were still mobile, in a prospective study of 80 DMD participants (94% on GC treatment) stratified into two groups by the *SPP1* genotype (TT vs. TG/GG), other motor function tests were studied. Achievements in terms of NSAA and 6MWT were followed for 12 months, and the G allele subgroup showed a significant decline in functional outcome [24]. In addition to the effect on motor functions, favorable effects of the *SPP1* rs28357094 genotype were reported for GC treatment, suggesting its potential as a pharmacodynamic biomarker of GC response [10,24,41,43]. In the Chinese cohort *SPP1*, the effect of rs28357094 was not studied as the frequency of G allele was lower than 0.01, but rs11730582 was introduced as a new genetic modifier of GC therapy tested on 326 patients [43]. Our study did not confirm the interaction of the rs28357094 *SPP1* genotype and GC therapy, which could be related to introduction of therapy later in life and the heterogeneity in therapy duration among patients.

The distribution of specific *LTBP4* haplotypes in our patients was similar to that seen in other studied cohorts, with VTTT and IAAM being the most common and accounting for more than 80% of haplotypes [27,41,44]. We did not observe an association of the IAAM haplotype with LoA, according to recessive models, as was the case in recent study conducted in Chinese DMD patients [43]. This association was not observed when stratifying patients according to the dominant and additive models either. On the other hand, a statistically significant effect was shown through many studies, starting with Flanigen and coworkers who first reported the *LTBP4* haplotype as a genetic modifier of DMD in the United Dystrophinopathy Project cohort composed of 254 participants [27]. IAAM haplotype, in a recessive model, was associated with a 1.5–2 year delay in median LoA. All four SNPs rs2303729, rs1131620, rs1051303, and rs10880 were associated with later LoA, but rs10880 showed the most significant single effect [27,45]. These results were confirmed in a multicenter study of 265 participants from five European centers, emphasizing the protective effect of the IAAM haplotype on LoA [41]. The effect of *LTBP4* was assessed in a multiethnic CINRG-DNHS cohort, but the whole cohort did not reach a statistically significant association of the IAAM haplotype with age at LoA. When the data were analyzed separately in a smaller subgroup of Caucasians, the recessive T genotype at rs10880 was found to be significantly associated with prolonged mobility, delaying the median age at LoA for 2.4 years [10]. On the contrary, in the Italian multicenter study, researchers aiming to assess the link between DMD genetic modifiers and dilated cardiomyopathy noted that the recessive IAAM haplotype was associated with earlier age at LoA (IAAM/IAAM 9.7 years vs. 10.8 years for other haplotypes and 11.1 years for VTTT/VTTT) [44]. In cohorts from four countries and five centers from van den Bergen, analyzing SNPs in the *SPP1* and *LTBP4* genes, the median delay at LoA was found to range from 10 to 18 months in favor of GC therapy [41].

In our cohort, we did not detect a significant effect of the T allele of rs1883832 in the *CD40* gene on LoA, according to our dominant model (TT and CT genotypes). As a modifier gene of DMD, *CD40* was described in 2016 in a genome-wide association study of exon variants in from 384 selected genes belonging to two pathways using an additive model [31]. Validation was performed in 660 participants from multiple independent DMD cohorts using both additive and dominant models, finding the statistically significant conclusion of a 1 year earlier median age at LoA in patients with the minor T allele. A study analyzing the effect of *CD40* rs1883832 genotype in DMD patients in two cohorts with meta-analysis on pulmonary function showed a negative effect of rs1883832 genotypes with T allele on lower FVC and earlier LoA [12].

Three examined DMD genetic modifiers are involved in the crosstalk of the NF-κB and TGF-β pathways, which plays a role in the pathogenesis of DMD. In addition, both osteopontin and LTBP4 directly modify sarcolemma repair in myofibers of normal and dystrophic muscles [46]. It can be assumed that the modifying effects of *SPP1*, *LPTB4,* and *CD40* variants are not independent, which may further shape their effect. We performed cluster analysis to describe the possible subgrouping of non-ambulant patients according to the genetic factors. Although we did not observe an association of an individual variant in common survival analyses, the profile of the two obtained clusters was consistent with the observed effects of variants in other DMD cohorts [21,27,31]. Importantly, the difference in age at LoA between clusters showed a statistical trend. The cluster showing a statistical trend toward a later LoA (cluster II) comprised fewer individuals carrying variants associated with a greater risk of earlier LoA. Specifically, almost all of these patients had at least one copy of the protective IAMM *LPTB4* haplotype. In addition, they were less likely to carry the T allele in the *CD40* gene and distal mutations in the *DMD* gene. Our results obtained from the cluster analysis imply that the effect of the examined DMD modifiers is probably compounded and not isolated, considering the fact that they are involved in the same pathological processes. Not reaching a statistically significant difference in LoA between clusters was expected due to the fact that the selected variants were not the only modifiers of DMD [45,47,48], in addition to the limitation related to sample size.

## 5. Conclusions

This is the first study on the genetic modifiers of DMD in a south Europe DMD population. The mutations in the *DMD* gene and allele frequencies of the SNP were distributed in a similar way to that seen in other European populations. We found that GC therapy significantly delayed the age of LoA. Patients not using GC therapy and carrying “distal” DMD mutations tended to lose mobility prior to patients with “proximal” mutations in the *DMD* gene. We did not show statistically significant effects of the selected genetic modifiers on the progression of the DMD using survival analyses. However, our results imply that cluster analysis may be able to identify patient subgroups carrying a combination of the genetic variants that modify LoA. We believe that bringing together patients from the southeastern Europe would provide better insight into the impact of the genetic modifiers of DMD and lead to a stronger conclusion on this topic for this part of Europe.

## Figures and Tables

**Figure 1 genes-13-01385-f001:**
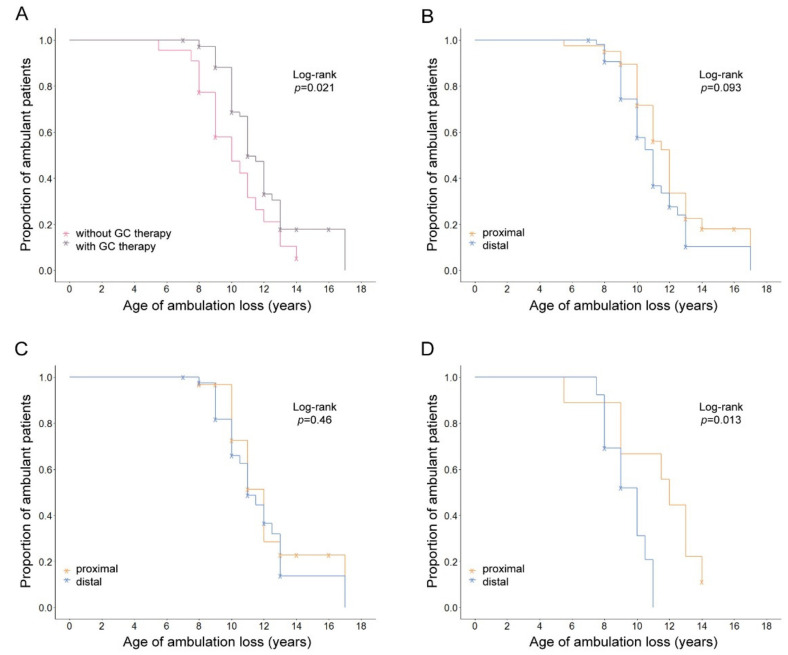
Kaplan–Meier plots showing the effect of GC therapy and mutation location on age at LoA for 95 patients with Duchenne muscular dystrophy. (**A**) The two survival lines represent patients stratified by GC therapy, where GC-treated patients had a later age at LoA compared to untreated patients. (**B**) The two survival lines represent patients stratified by mutation location within the *DMD* gene, where patients with the proximal location showed a trend of later age at LoA compared to patients with the distal location. The effect of mutation location was not significant in (**C**) patients with GC therapy, while, in (**D**) patients without GC therapy, proximal location was shown to have a protective effect. Log-rank test was used to compare different Kaplan–Meier curves, and the corresponding *p*-values are shown on the top right corner of all plots. Censored patients are indicated with a cross on their survival lines.

**Table 1 genes-13-01385-t001:** Mean age at loss of ambulation (LoA) in DMD patients, stratified by glucocorticoid (GC) therapy, *DMD* mutation location, *SPP1* and *CD40* genotypes, and *LPTB4* haplotypes.

	GC Therapy	*N* (Events)	Mean Age of LoA Years (95% CI)	KM Log-Rank *p*
	Yes	73 (45)	11.14 (10.52–11.76)	0.021
	No	22 (19)	9.95 (8.9–11)	
*DMD* mutation location				
Proximal	/	40 (26)	11.08 (10.21–11.95)	0.093
Distal		55 (38)	10.59 (9.88–11.3)	
Proximal	Yes	31 (18)	11.17 (10.25–12.09)	0.46
Distal		42 (27)	11.13 (10.24–12.02)	
Proximal	No	9 (8)	10.88 (8.51–13.25)	0.013
Distal		13 (11)	9.27 (8.4–10.14)	
*SPP1* (rs28357094)		93 (62)		
TT	/	55 (36)	10.94 (10.23–11.65)	0.32
TG + GG		38 (26)	10.62 (9.68–11.56)	
TT	Yes	46 (28)	11.02 (10.19–11.85)	0.79
TG + GG		25 (15)	11.5 (10.35–12.65)	
TT	No	9 (8)	10.69 (8.95–12.43)	0.69
TG + GG		13 (11)	9.41 (7.95–10.87)	
*CD40* (rs1883832)		95 (64)		
CC	/	40 (31)	10.69 (9.94–11.44)	0.24
CT + TT		55 (33)	10.88 (10.07–11.69)	
CC	Yes	27 (20)	10.95 (9.99–11.91)	0.45
CT + TT		46 (25)	11.3 (10.42–12.18)	
CC	No	13 (11)	10.23 (8.83–11.63)	0.8
CT + TT		9 (8)	9.56 (7.57–11.55)	
*LTBP4* (Haplotype)		89 (58)		
Other/other	/	41 (27)	10.39 (9.5–11.28)	0.61
Other/IAAM + IAAM/IAAM		48 (31)	10.69 (10.12–11.26)	
Other/other	Yes	34 (22)	11.02 (10.16–11.88)	0.43
Other/IAAM + IAAM/IAAM		33 (17)	10.62 (9.9–11.34)	
Other/other	No	7 (5)	7.6 (5.99–9.21)	0.43
Other/IAAM + IAAM/IAAM		15 (14)	10.79 (9.77–11.81)	
Other/other	/	41 (27)	10.39 (9.5–11.28)	0.56
Other/IAAM		40 (26)	10.54 (9.91–11.17)	
IAAM/IAAM		8 (5)	11.5 (9.74–13.26)	
Other/other	Yes	34 (22)	11.02 (10.16–11.88)	0.62
Other/IAAM		28 (15)	10.63 (9.8–11.46)	
IAAM/IAAM		5 (2)	10.5 (4.15–16.85)	
Other/other	No	7 (5)	7.6 (5.99–9.21)	0.54
Other/IAAM		12 (11)	10.41 (9.26–11.56)	
IAAM/IAAM		3 (3)	12.17 (8.58–15.76)	
Other/other + other/IAAM	/	81 (53)	10.46 (9.93–10.99)	0.29
IAAM/IAAM		8 (5)	11.5 (9.74–13.26)	
Other/other + other/IAAM	Yes	62 (37)	10.86 (10.27–11.45)	0.43
IAAM/IAAM		5 (2)	10.5 (4.15–16.85)	
Other/other + other/IAAM	No	19 (16)	9.53 (8.44–10.62)	0.34
IAAM/IAAM		3 (3)	12.17 (8.58–15.76)	

**Table 2 genes-13-01385-t002:** Effect of glucocorticoid (GC) therapy, mutation location in the *DMD* gene, and *SPP1*, *CD40,* and *LPTB4* genes on age at loss of ambulation (LoA).

	N (Events)	HR (95% CI)	Z-Score	p-Value	Cox p-Value
GC therapy					0.068
Nontreated	22 (19)	1		0.01	
Treated	73 (45)	0.44 (0.23–0.83)	−2.51		
DMD mutation location					
Proximal	40 (26)	1		0.03	
Distal	55 (38)	1.92 (1.07–3.47)	2.18		
SPP1 (rs28357094)	93 (62)				
TT	55 (36)	1		0.9	
TG + GG	38 (26)	1.03 (0.60–1.78)	0.12		
CD40 (rs1883832)	95 (64)				
CC	40 (31)	1		0.59	
CT + TT	55 (33)	0.86 (0.5–1.48)	−0.54		
LTBP4 (Haplotype)	89 (58)				
Other/other	41 (27)	1		0.25	
Other/IAAM + IAAM/IAAM	48 (31)	0.72 (0.41–1.26)	−1.15		
Interaction					
GC treatment × DMD mutation (distal)	42 (27)	0.21 (0.06–0.74)	−2.42	0.02	0.056
GC treatment × SPP1 rs28357094 (TG + GG)	25 (15)	0.69 (0.19–2.55)	−0.56	0.58	
GC treatment × CD40 rs1883832 (CT + TT)	46 (25)	0.68 (0.19–2.38)	−0.6	0.55	
GC treatment × LTBP4 (other/IAAM + IAAM/IAAM)	33 (17)	1.25 (0.29–5.4)	0.3	0.76	

**Table 3 genes-13-01385-t003:** Cluster profiles of non-ambulant DMD patients according to genetic factors and age at loss of ambulation (LoA).

Variable	Description	Cluster I	Cluster II	*p*-Value
		*N* = 41	*N* = 23	
*SPP1* rs28357094 *	TT	24	14	1 **
	TG + GG	17	9	
*CD40* rs1883832 *	CC	13	18	5.7 × 10^−4^ **
	CT + TT	28	5	
*LTBP4* haplotypes *	Other/other	26	1	2.8 × 10^−6^ **
	Other/IAAM + IAAM/IAAM	15	22	
Location of mutation	Proximal	12	14	0.018 **
	Distal	29	9	
Age at LoA	≤10.75 years	24	8	0.06 ***
	>10.75 years	17	15	

* Dominant effect of less frequent allele/haplotype. ** Fischer’s exact test. *** Wilcoxon–Mann–Whitney test.

## Data Availability

The data presented in the study are available on request from the corresponding author due to restrictions (privacy).

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
