# Peer review of "LTBP4, SPP1, and CD40 Variants: Genetic Modifiers of Duchenne Muscular Dystrophy Analyzed in Serbian Patients"

_genes, 2022, doi:10.3390/genes13081385_

Round 1

Reviewer 1 Report

This is a well conducted and clearly structured manuscript, analyzing the effect of some known DMD genetic modifiers on a cohort of 95 molecularly confirmed DMD patients. The methods are scientifically sound, and the use of hierarchical clustering is a clever addition to the standard Cox-regression analysis of loss of ambulation, especially with a small-sized sample. The findings, both the positive and negative ones, are well discussed, and exhaustively compared to the existing literature.

Suggestions:

The LTBP4 haplotype was studied with either recessive or dominant models; an alternative, especially in Cox regression, would be to apply an additive model, which is “in between” dominant and recessive. The authors may consider adding this analysis, especially in the Cox regression part (rather than the hierarchical clustering which works better with dichotomic variables).

Also, the authors may include some data about del 3-7 and del 45, even if they do not include them in the overall study (which may have been done with a dedicted covariate). It would be interesting to know what LoA looked like in these patients, although the numbers will probably be small.

Minor edits:

Line 75: overexpresed  overxpressed

Lines 95-98: mice do not have the VTTT/IAAM haplotype, but rather a 36 bp insertion/deletion polymorphism.

Line 265: recesive  recessive

Table 3: Fischer  Fisher

Line 361 and elsewhere: gene modifiers  genetic modifiers (“gene modifiers” sounds like something that modifies a gene; “genetic modifiers” seems more appropriate for variants which modify a phenotype)

Line 364: well characterized 95 unrelated patients  95 well characterized, unrelated patients

Line 404: our  ours

Line 424: Flaningen  Flanigan

Line 457: we did not observed - we did not observe

Finally, I suggest to review the overall manuscript with a native speaker, especially for punctuation and the use of articles (“a”, “the”). This may be done at a proofing stage.

Reviewer 2 Report

Overall Comments: The manuscript by Kosac and colleagues centers on the identification and characterization of LTBP4, SPP1, and CD40 genetic modifiers in a Duchenne muscular dystrophy (DMD) Serbian population cohort. The authors assessed a cohort of 95 DMD patients from a Serbian registry, with inclusion criteria for DMD and additional criteria of consideration DMD genetic mutation, age, and glucocorticoid status. Previously validated genetic modifiers LTBP4, SPP1, and CD40 that affect DMD patient ambulatory status were assessed for their impact on mobility and ambulation. The manuscript is relatively straight-forward experimentally; however, there are a large number of grammar and coherence issues sprinkled in the introduction and discussion. Additionally, there are some questions with regards to the stratifying of DMD mutation status as “proximal” versus “distal” as more justification for the lack of inclusion over exact DMD mutation is warranted. These questions along with some of the lacking background are needed to strengthen the overall findings of this manuscript.

Major Comments:

1. While reporting the DMD mutations as “proximal” or “distal” is somewhat informative (Table 1), it would be more informative to know either the type of DMD mutations (e.g. exon-deletions, missense, microdeletion, etc.) and/or the actual DMD mutation frequencies (e.g. exons 45-55, DMD hot-spot region), to more accurately assess the impact of the genomic modifier variants.

2. Another question that arises is how many of these 95 DMD patients assessed were from de novo inheritance versus female (e.g. mother) carrier mutations? This part would have implications with regards to cross comparisons of the variant allele transmission frequencies.

3. The biggest question I had with this study, was what was the frequency of multiple genetic modifiers in this DMD population? E.g. what percent had the SPP1 rs28357094 and CD40 rs1883832 SNP alleles?

4. Minor comment. I would add an Oxford comma in the title to make it read: “LTBP4, SPP1, and CD40 variants – genetic modifiers of Duchenne muscular dystrophy, analyzed in Serbian patients”. The comma after SPP1 is essential to delineate the individual modifiers.

5. Page 2, line 42. Shouldn’t this be “leading”, not “lead”.

6. Page 2, Line 44 The statement “Cardiomyopathy and respiratory failure are evolving”…this is a misleading statement. “progressing” is a better term.

7. Page 3, Line 101. Needs a space for “Tcells”.

8. Page 12, Line 374. The reference to the “McDonald’s study” is awkward as it is not a study named after this particular investigator. The wording of this reference and sentence should be changed.

9. Page 13, Line 441. Please clarify the “dominant model”? This sentence is unclear. “In our cohort, we did not detect significant effect of T allele of rs1883832 in CD40 gene on 441 LoA, according to dominant model.”.

Round 2

Reviewer 2 Report

The authors have made significant clarifications with regards to methodology and analysis of the respective DMD Serbian cohorts that were analyzed. The experimental methods and findings are overall sound. They have made improvements in the grammar and English clarity. There's a bit of refinement in this area that can be made additionally with work between the handling editor and authors. Otherwise, I have no additional concerns.